



# Clustering diurnal cycles of day-to-day temperature change to understand their impacts on air quality forecasting in mountain-basin areas

Debing Kong[1,2], Shigong Wang[3,4], Guicai Ning[3,5*], Jing Cong[6], Ming Luo[5,7], Xiang Ni[1,2], Mingguo Ma[1,2]

[1]Chongqing Jinfo Mountain Karst Ecosystem National Observation and Research Station, School of Geographical Sciences, Southwest University, Chongqing, 400715, China
[2]Chongqing Engineering Research Center for Remote Sensing Big Data Application, School of Geographical Sciences, Southwest University, Chongqing, 400715, China
[3]The Gansu Key Laboratory of Arid Climate Change and Reducing Disaster, College of Atmospheric Sciences, Lanzhou University, Lanzhou 730000, China
[4]Sichuan Key Laboratory for Plateau Atmosphere and Environment, School of Atmospheric Sciences, Chengdu University of Information Technology, Chengdu 610225, China
[5]Institute of Environment, Energy and Sustainability, The Chinese University of Hong Kong, Shatin, N.T., Hong Kong, China
[6]Tianjin Municipal Meteorological Observatory, Tianjin 300074, China
[7]School of Geography and Planning, and Guangdong Key Laboratory for Urbanization and Geo-simulation, Sun Yat-sen University, Guangzhou 510275, China

*Correspondence to: Dr. Guicai Ning (ninggc09@lzu.edu.cn)

**Abstract.** Air pollution is substantially modulated by meteorological conditions, and especially their diurnal variations may play a key role in air quality evolution. However, the behaviors of temperature diurnal cycles along with the associated atmospheric condition and their effects on air quality in China remain poorly understood. Here, for the first time we examine the diurnal cycles of day-to-day temperature change and reveal their impacts on winter air quality forecasting in mountain-basin areas. Three different diurnal cycles of the preceding day-to-day temperature change are identified and exhibit notably distinct effects on the day-to-day changes in atmospheric dispersion conditions and air quality. The diurnal cycle with increasing temperature obviously enhances the atmospheric stability in the lower troposphere and suppresses the development of the planetary boundary layer, thus deteriorating the air quality on the following day. By contrast, the diurnal cycle with decreasing temperature in the morning is accompanied by a worse dispersion condition with more stable atmosphere stratification and weaker surface wind speed, thereby substantially worsening the air quality. Conversely, the diurnal cycle with decreasing temperature in the afternoon seems to improve air quality on the following day by enhancing the atmospheric dispersion conditions on the following day. The findings reported here are critical to improve the understanding of air pollution in mountain-basin areas and exhibit promising potential for air quality forecasting.





## 1. Introduction


Air pollution is not only affected by anthropogenic emissions (Streets et al., 2001; Zhang et al., 2009; Kelly and Zhu, 2016),
but also controlled by atmospheric dispersion conditions (Wei et al., 2011; Li et al., 2015; Ye et al., 2016; Zhang et al.,
2020). Stagnant meteorological conditions significantly contribute to the formation and maintenance of heavy air pollution
as they play important roles in regulating the increment of air pollutants concentrations (Deng et al., 2014; Bei et al., 2016;
Zhang et al., 2016; Wang et al., 2018). It is noted that atmospheric dispersion capacity is substantially modulated by synoptic
patterns and hence the evolutions of large-scale synoptic systems can lead to the improvement or deterioration of air quality
(Yarnal, 1993; Miao et al., 2017; Ning et al., 2019; Dong et al., 2020; Ning et al., 2020). In China, high anthropogenic
emissions from coal-fired heating (Xiao et al., 2015), frequent temperature inversion (Xu et al., 2019; Feng et al., 2020; Guo
et al., 2020), and shallow planetary boundary layer (PBL) structure (Li et al., 2017; Miao et al., 2018; Su et al., 2020) result
in frequent occurrence of heavy air pollution events in winter. These factors highlight the significance of further revealing
the physical mechanism of atmospheric dispersion evolutions.

The behaviors of diurnal cycles of atmospheric dispersion conditions and their effects on air quality remain poorly
understood despite air pollution significantly modulated by atmospheric dispersion conditions has been well demonstrated.
For instance, as a typical synoptic process occurring in winter in China, the cooling process could cause rapid changes in
meteorological and environmental conditions. Cooling processes induce significant day-to-day temperature variations and
thus result in substantial changes in air quality (Hu et al., 2018; Ning et al., 2018b; Kang et al., 2019). Many previous studies
revealed that cooling processes could remove air pollutants by invading lots of cold fresh airflows (Kalkstein and Corrigan,
1986; Gimson, 1994; Hu et al., 2018; Ning et al., 2018b) or exacerbate air pollution by transporting air pollutants (Fu et al.,
2008; Ding et al., 2013; Luo et al., 2018; Kang et al., 2019). Nevertheless, most of these studies did not consider the
influences of diurnal cycles of cooling processes on air quality. Are the influences of cooling processes occurring during
daytime and nighttime on air quality similar or different? The key questions include what are the behaviors of the diurnal
cycles of atmospheric dispersion conditions and how these behaviors affect air quality, especially how the diurnal cycles of
day-to-day temperature change affect air pollution. Exploring the answers to these questions is critical for fully
understanding of winter air pollution and is also urgently needed for improving air quality forecasting in China.

Sichuan Basin (SCB) is one of the heaviest air pollution areas in China (Zhang et al., 2012; Ning et al., 2018a). With a high
population density in SCB, its heavy air pollution thus poses serious health hazards to local residents (Liao et al., 2017; Qiu
et al., 2018; Zhu et al., 2018; Zhao et al., 2018). It is noted that SCB has a unique topography, with Qinling-Daba and Wu
mountains in the north and east and with Qinghai-Tibet Plateau and Yunnan-Guizhou Plateau in the west and south of the
basin (**Fig. 1**). The combination of these complex topography results in unique weather and climate, like the southwest
vortex and the Huaxi Autumn rain season etc. The southwest vortex, southern branch, and Qinghai-Tibet high pressure are





often formed over SCB or Tibetan plateau and the complex synoptic systems significantly affect atmospheric dispersion
conditions (Wang et al., 1993; Wei et al., 2014; Feng et al., 2016; Yu et al., 2016; Ning et al., 2019; Ning et al., 2020).
Therefore, both the physical mechanism of atmospheric conditions effects on air pollution and the air quality forecasting in
SCB are more complicated than these in the eastern plain regions of China (Chen and Xie, 2012; Wang et al., 2014; Ning et
al., 2019; Zhang et al., 2019). To better understand the formation mechanism of air pollution and improve air quality
forecasting in mountain-basin areas, the effects of diurnal variations of atmospheric dispersion conditions on winter air
quality in SCB call for urgent examinations.

The scientific goals of this study are to first cluster the typical diurnal cycles of day-to-day temperature change in SCB
during wintertime and then to examine the mechanisms underlying the effects of the identified typical diurnal cycles on the
following day-to-day air quality changes. Our study is expected to better understand the physical mechanism of air quality
evolutions and improve air pollution forecasting in mountain-basin areas. The rest of this paper is organized as below. Data
and methodology are introduced in section 2. Section 3 describes the results of our study. Discussion related to our findings
is given in section 4. Our conclusions are summarized in section 5.
**2. Data and methodology**
**2.1 Air quality data**
Hourly concentrations of surface $PM_{2.5}$ (particulate matter with an aerodynamic diameter equal to or less than 2.5 μm), $PM_{10}$
(particulate matter with an aerodynamic diameter equal to or less than 10 μm), $SO_2$ (sulfur dioxide), $NO_2$ (nitrogen dioxide),
and CO (carbon monoxide) in the winters (December–February) from December, 2014 to February, 2020 in 18 cities of SCB
(**Fig. 1**) are obtained from the Ministry of Ecology and Environment of the People's Republic of China
(http://www.mee.gov.cn/xxgk2018/). We calculate the city-wide average concentrations of the five air pollutants by
arithmetically averaging their concentration at the national air quality monitoring sites located in the urban areas of that city,
based on the technical regulation for ambient air quality assessment (on trial) (MEP, 2013; Ning et al., 2020). Among the 18
cities in SCB, ten (Leshan, Meishan, Ziyang, Guangyuan, Bazhong, Ya'an, Dazhou, Suining, Guangan, and Neijiang) began
monitoring air quality since January 1, 2015. Hence, the starting date of air quality data for these 10 cities is December 1,
2015. The starting date of air quality data for the rest 8 cities (Chengdu, Deyang, Mianyang, Zigong, Yibin, Luzhou,
Nanchong and Chongqing) is December 1, 2014.
**2.2 Meteorological observational data**
Hourly winter surface temperature data observed at 105 meteorological stations in SCB (**Fig. 1**) from December 2006 to
February 2020 are also collected. Their regional averages are used to determine the diurnal cycles of day-to-day temperature
change. Additionally, daily mean surface wind speed in the 18 cities of SCB is also collected. To explore the thermodynamic



structure of the lower troposphere, daily potential temperature profiles at 20:00 Beijing time (BJT, UTC+8 h) from four
sounding stations in SCB are also obtained. Four sounding stations, including Chengdu, Yibin, Dazhou, and Chongqing, are
located in the northwest, southwest, northeast and southeast of the basin, respectively (See the orange dots in **Fig.1**). All
these surface meteorological observations are obtained from the China Meteorological Administration (CMA)
(http://data.cma.cn/data/).
**2.3 ERA-5 reanalysis data**
To obtain winter lower troposphere stability, 700 hPa temperature and air pressure and air temperature at 2 m above the
ground from December 2014 to February 2020 are collected from daily ERA-5 reanalysis data (0.25°×0.25° grids)
(https://cds.climate.copernicus.eu/cdsapp#!/dataset). We collect the reanalysis data at four times each day (UTC 00:00,
06:00, 12:00 and 18:00) to calculate their daily mean values. The PBL height (PBLH) data at UTC 06:00 (14:00 BJT) are
also obtained. PBLH is defined as the lowest model level where the bulk Richardson number first reaches the threshold value
of 0.25 (Beljaars, 2006).
**2.4 Quantitative measurements of meteorological and air quality variables**
**2.4.1 Lower troposphere stability**
The lower troposphere stability (LTS) is defined as the differences in potential temperature between 700 hPa and the surface
(Slingo, 1987). LTS can describe the thermal state of the lower troposphere and thus can be used to evaluate the vertical
mixing of air pollutants in the lower troposphere (Guo et al., 2016a; Guo et al., 2016b). A larger LTS indicates a stronger
stability in the lower troposphere and a weaker vertical mixing of air pollutants.
**2.4.2 Day-to-day changes in meteorological conditions and air quality**
The day-to-day temperature change for each hour of a given day is defined by the hourly temperature differences between
two neighboring days (Karl et al., 1995):
$\triangle T=T_i-T_{i-1}$        (1)
where $\triangle T$ refers to day-to-day temperature change, $T_i$ and $T_{i-1}$ are the hourly temperatures at the specific time of the day and
the previous day, respectively.

To investigate the effects of diurnal cycles of day-to-day temperature change on air quality, we also calculate the day-to-day
changes in air pollutants concentrations and atmospheric dispersion conditions following the temperature change within one
day. The following day-to-day changes in air pollutants concentrations (or atmospheric dispersion conditions) are defined by
the differences in air pollutants concentrations (or meteorological conditions) between the next day and the current day:
$\triangle PC=PC_{i+1}-PC_i$        (2)



where *PC* represents PBLH, LTS, vertical potential temperature profiles (PT), surface wind speed (WS), or the
concentrations of $PM_{2.5}$, $PM_{10}$, $SO_2$, $NO_2$, and CO. $\Delta PC$ represents the following day-to-day changes in PBLH, LTS, PT,
WS, and five air pollutants concentrations. $PC_{i+1}$ is the daily mean LTS, WS, and air pollutants concentrations, or the PBLH
at 14:00 BJT and PT at 20:00 BJT on the next day. $PC_i$ is the daily mean LTS, WS, and air pollutants concentrations, or the
PBLH at 14:00 BJT and PT at 20:00 BJT on the current day.
**2.5 K-means clustering**
Clustering methods divide the objects into specific groups, with the goal that all data objects assigned to the same cluster
have common characteristics while different clusters have distinct characteristics (Darby, 2005). The clustering methods
have been widely used in climate and environmental researches (Bardossy et al., 1995; Cavazos, 2000; Luo and Lau, 2017;
Bernier et al., 2019). In this study, the regional average values of day-to-day temperature change in SCB and the K-means
clustering method (MacQueen, 1967) are selected to classify the diurnal cycles of day-to-day temperature change, because of
the simplicity and convergence characteristics of K-means clustering method. The details of K-means clustering method can
refer to MacQueen (1967) and (Mokdad and Haddad, 2017). Additionally, the Calinski-Harabasz criterion, also known as the
variance ratio criterion, is utilized to determine the optimal number of clusters (Caliński and Harabasz, 1974). The ultimate
goal of Calinski-Harabasz criterion is to maximize the variance measure ratio of homogeneity within a cluster and
heterogeneity between clusters (Chikumbo and Granville, 2019).

**3. Results**
**3.1 Diurnal cycles of day-to-day temperature change**
The selection of optimal number of clusters is illustrated in **Fig. 2**, which shows Calinski-Harabasz values associated with
the numbers of clusters ranging from two to ten. The Calinski-Harabasz value with three clusters reaches the highest value,
indicating that the optimal number of clustering is three. Three dominant diurnal cycles of day-to-day temperature change
are therefore identified in SCB. The three typical diurnal cycles of day-to-day temperature change are depicted in **Fig. 3**. The
days for *Cluster* 1, *Cluster* 2, and *Cluster* 3 are 455 (accounting for 36.9 % of total days), 413 (33.5%), and 365 days
(29.6%), respectively, indicating that the differences in the occurrence frequency among the three diurnal cycles are not
noticeable. However, the diurnal cycles of day-to-day temperature change among the three clusters exhibit obvious
differences.

In particular, *Cluster* 1 (diurnal cycle with increasing temperature), all the temperature changes are positive for 24 hours
throughout all day, indicating that temperature increases during the past 24-hour and exhibits a maximum change
approaching 1.5 ℃ between 16:00 BJT and 17:00 BJT. *Cluster* 2 (diurnal cycle with decreasing temperature in the





afternoon), the temperature changes show negative values after 12:00 BJT and drop to trough between 16:00 BJT and 17:00
BJT with the minimum value of -1.5 ℃, indicating that the cooling process is obvious in the afternoon. After 17:00 BJT, the
absolute values of temperature change begin to decrease. The most prominent feature of *Cluster* 2 is that the obvious
decrease in temperature appears in the afternoon. *Cluster* 3 (diurnal cycle with decreasing temperature in the morning), all
temperature changes are negative for 24 hours throughout all day, and the obviously cooling process appears from 00:00 BJT
to 09:00 BJT. The temperature changes show the minimum value approaching -1.5 ℃ between 07:00 BJT and 09:00 BJT.
After 09:00 BJT, the absolute values of temperature change gradually reduce and are nearly close to zero in the afternoon.
The most prominent feature of *Cluster* 3 is that the obvious decrease in temperature appears in the morning.
**3.2 Air quality in relation to the identified diurnal cycles**
Heavy air pollution during winter in SCB is mainly caused by high concentrations of particulate matter ($PM_{2.5}$ and $PM_{10}$)
(Ning et al., 2018a). Therefore, the day-to-day changes in $PM_{2.5}$ and $PM_{10}$ concentrations following the three identified
diurnal cycles within one day are investigated. **Fig. 4** depicts the spatial distributions of the following day-to-day changes in
$PM_{2.5}$ and $PM_{10}$ concentrations associated with the three typical diurnal cycles. Under the diurnal cycle with increasing
temperature (*Cluster* 1), nearly all parts of SCB experience increases in $PM_{2.5}$ and $PM_{10}$ concentrations on the following day
(**Fig. 4a** and **d**). The regional average changes in $PM_{2.5}$ and $PM_{10}$ concentrations are up to +3.95 µg/m$^3$ and +5.89 µg/m$^3$,
respectively.

On the contrary, negative changes in $PM_{2.5}$ and $PM_{10}$ concentrations are observed in the entire basin for the diurnal cycle
with decreasing temperature in the afternoon (*Cluster* 2) (**Fig. 4b** and **e**), indicating the improvement of air quality on the
following day. The regional average changes in $PM_{2.5}$ and $PM_{10}$ concentrations are up to -8.93 µg/m$^3$ and -11.50 µg/m$^3$,
respectively. Under the diurnal cycle with decreasing temperature in the morning (*Cluster* 3), all parts of SCB experience
increases in $PM_{2.5}$ and $PM_{10}$ concentrations (**Fig. 4c** and **f**), indicating the deterioration of air quality on the following day. It
is noted that opposite changes in $PM_{2.5}$ and $PM_{10}$ concentrations are observed between *Cluster* 3 and *Cluster* 2 even though
both of the two diurnal cycles show decreasing temperature. Compared with the diurnal cycle with increasing temperature
(*Cluster* 1), the increases in $PM_{2.5}$ and $PM_{10}$ concentrations are larger for *Cluster* 3, and the regional average changes in
$PM_{2.5}$ and $PM_{10}$ concentrations are up to +5.36 µg/m$^3$ and +5.91 µg/m$^3$, respectively.

The contributions of gaseous pollutants in SCB to winter air pollution are also very important as SCB has a large number of
motor vehicles and industries (Ning et al., 2018a). Therefore, the following day-to-day changes in three major gaseous ($SO_2$,
$NO_2$, and CO) concentrations associated with the three diurnal cycles are also investigated. Similar to particulate matter, the
relationships between the following day-to-day changes in gaseous pollutants concentrations and the three diurnal cycles are
consistent with the results showed in **Fig. 4**. As shown in **Fig. 5**, nearly all parts of SCB experience increases in $SO_2$, $NO_2$,
and CO concentrations on the following day for *Cluster* 1 (diurnal cycle with increasing temperature) and *Cluster* 3 (diurnal





cycle with decreasing temperature in the morning). On the contrary, negative changes in $SO_2$, $NO_2$, and CO concentrations
are observed in the entire basin for *Cluster* 2 (diurnal cycle with decreasing temperature in the afternoon).

**Figs. 4** and **5** collectively indicate that the air quality in SCB corresponding to *Custer* 1 and *Cluster* 3 will deteriorate on the
following day, while the air quality corresponding to *Cluster* 2 will improve. These results suggest that the modulations of
diurnal cycles of day-to-day temperature change on the following day-to-day changes in winter air quality are obvious and
important. Thus, the diurnal cycles of day-to-day temperature change exhibit promising potential for winter air quality
forecasting on the following day in SCB.
**3.3 Mechanism of the identified diurnal cycles effects on air quality**
To reveal the potential influence mechanism of the diurnal cycles of day-to-day temperature change on the following day-to-
day changes in air quality, the atmospheric dispersion conditions corresponding to the three identified diurnal cycles are
investigated. Firstly, the following day-to-day changes in PT vertical profiles at four sounding stations in SCB (**Fig. 6**) are
examined to explore the thermodynamic structure in the lower troposphere. Then, the following day-to-day changes of the
three meteorological parameters related to atmospheric dispersion conditions, including LTS (**Fig. 7a–c**), PBLH (**Fig. 7d–f**),
and WS (**Fig. 7g–i**) are also investigated to evaluate the evolutions of atmospheric dispersion capacity.

Under the diurnal cycle with increasing temperature (*Cluster* 1), three sounding stations (Yibin, Dazhou, and Chongqing)
experience increases in PT between 950 hPa to 800 hPa on the following day (**Fig. 6d**, **g**, and **j**). In Chengdu, decreased PT
is observed below 900 hPa, while increased PT appears between 900 hPa to 750 hPa (**Fig.6a**). All the PT profiles over the
four sounding stations show higher temperature change in the level between middle level (800-850 hPa) than the lower level
(900-950 hPa), which could enhance the atmospheric stability in the lower troposphere. As shown in **Fig.7a**, increased LTS
are observed in most of the cities in SCB, indicating the atmospheric stratification in the lower troposphere becomes more
stable. The stable atmospheric stratification inhibits the vertical mixing of the atmosphere and suppresses the development of
PBL (Karppinen et al., 2001; Bei et al., 2016). As shown in **Fig. 7d**, obviously decreased PBLH are observed in all 18 cities
of SCB.

Additionally, we also analyzed the following day-to-day changes in surface wind speed as the wind speed can represent the
horizontal dispersion capacity of air pollutants (Lu et al., 2012; Deng et al., 2014). No noticeable decreases in wind speed
appear in SCB (**Fig. 7g**). These results suggest that the diurnal cycle with increasing temperature (*Cluster* 1) enhances
atmospheric stability in the lower troposphere, which can weaken the vertical exchange of airflow and then suppress the
development of PBL, resulting in a small dispersion space of air pollutants and poor air quality in SCB on the following day.
Compared with *Cluster* 1, opposite vertical structure of PT changes (**Fig. 6b**, **e**, **h**, and **k**) is observed for the diurnal cycle
with decreasing temperature in the afternoon (*Cluster* 2), which could weaken the atmospheric stability in the lower





troposphere. As shown in **Fig. 7b**, negative changes in LTS appear in all parts of SCB, enhancing the vertical exchange of
airflow and facilitating the development of PBL. As a result, increased PBLH is observed in all parts of SCB (**Fig. 7e**), and
the regional average increment is up to 93.0 m. At the same time, the weakened atmospheric stability in the lower
troposphere is also conducive to the development of surface wind speed. As shown in **Fig. 7h**, the surface wind speed in the
entire SCB is strengthened obviously, indicating the horizontal dispersion capacity of air pollutants is also improved. These
results suggest that the diurnal cycle with decreasing temperature in the afternoon weakens atmospheric stability in the lower
troposphere and creates good vertical mixing of airflow, which can promote the development of PBL and surface wind
speed, facilitating the improvement of air quality on the following day.

For the Cluster 3, the PT changes are not noticeable below 850 hPa over the four sounding stations. As shown in **Fig. 6c**, **f**, **i**,
and **l**, decreased PT is observed between 850 hPa and 700 hPa, while obviously increased PT appears above 700 hPa. This
vertical structure of PT changes suggests that the atmospheric stability is enhanced above PBL over SCB, which is
demonstrated playing key role in the formation of winter heavy air pollution events in the basin (Ning et al., 2018b; Ning et
al., 2019). As shown in **Fig. 7c**, increased LTS appears in the entire SCB, and the increments of LTS are obviously larger
than those for *Cluster* 1 (**Fig. 7a**), inhibiting the vertical mixing of atmosphere and suppressing the development of PBL. As
a result, decreased PBLH is observed in all parts of SCB. Compared with *Cluster* 1, the enhanced atmospheric stability
above PBL also suppresses the development of surface wind speed. As shown in **Fig. 7i**, all parts of SCB experience
decreases in surface wind speed, weakening the horizontal dispersion capacity of air pollutants. These results suggest that
both the vertical and horizontal dispersion capacity of air pollutants corresponding to *Cluster* 3 are worse than those
corresponding to *Cluster* 1. The differences in the atmospheric dispersion conditions between *Cluster* 3 and *Cluster* 1 can
explain well that the air quality deterioration is more serious for *Cluster* 3 than *Cluster* 1 (**Fig. 4** and **Fig. 5**).
**4. Discussion**
It's worth noting that the following day-to-day air quality changes between *Cluster* 2 and *Cluster* 3 in mountain-basin areas
are opposite, even though both of the two diurnal cycles are associated with cooling processes. In the cases of the cooling
process mainly occurring in the afternoon (*Cluster* 2), the atmospheric dispersion conditions are obviously improved,
resulting in air quality improvement on the following day. On the contrary, the atmospheric dispersion conditions are
obviously inhibited when the cooling process mainly appears in the morning (*Cluster* 3), resulting in air quality deterioration
on the following day. These findings could improve our understanding of the effects of cooling processes on air quality
(Kalkstein and Corrigan, 1986; Gimson, 1994; Hu et al., 2018; Ning et al., 2018b; Kang et al., 2019) and suggest that
comprehensive investigations for the effects of diurnal cycles of atmospheric dispersion conditions on air quality are
urgently needed in the future to fully understand the physical mechanism of air quality evolutions.





Additionally, both *Cluster* 1 and *Cluster* 3 are associated with weakened atmospheric dispersion conditions and lead to air
quality deterioration on the following day. However, obvious differences in PT vertical profiles (**Fig. 6**) between *Cluster* 1
and *Cluster* 3 are observed. Especially for *Cluster* 3, decreased PT is observed between 850 hPa and 700 hPa, while
obviously increased PT appears above 700 hPa (**Fig. 6c**, **f**, **i**, and **l**). This special vertical structure of PT is closely related to
the foehn that is formed under the synergistic effects of cooling processes and the Tibetan Plateau (Ning et al., 2019),
indicating a stable layer exits above PBL and acts as a lid covering the PBL (Ning et al., 2018b; Ning et al., 2019). The
vertical structure of PT are demonstrated playing key roles in the formation of winter heavy air pollution events in mountain-
basin areas by inhibiting the development of secondary circulation and PBL (Ning et al., 2018b; Ning et al., 2019). These
features suggest that the physical processes related to air pollution are more complex in mountain-basin areas than in the
areas with flat terrain and urgently need to be further explored in the future.

Our study highlights that the following day-to-day air quality changes in mountain-basin areas are notably affected by the
diurnal cycles of day-to-day temperature changes. We find that the identified diurnal cycles of day-to-day temperature
variation in our study can explain well the evolutions of atmospheric dispersion conditions and air quality on the following
day and thus could be useful for air quality forecasting in mountain-basin areas. Currently, numerical models (including
WRF-Chem model and CMAQ model) (Grell et al., 2005; Byun and Ching, 1999) and statistical models (including statistical
analysis, machine learning, and the hybrid linear–nonlinear method, etc.) (Huang, 1992; Chelani and Devotta, 2006; Borse,
2020) are the two typical methods that have been widely used to forecast air quality by combining weather conditions and
emission sources (Gidhagen et al., 2005). In the future, our findings should therefore be combined with numerical models or
statistical models to improve air quality forecasting in mountain-basin areas.
**5. Conclusions**
Taking SCB as an example, this study is the first examination of the behaviors of diurnal cycles of day-to-day temperature
change using hourly temperature observations and their effects on the following day-to-day air quality changes in mountain-
basin areas. Three diurnal cycles of day-to-day temperature change are identified, which notably affect the following day-to-
day air quality changes. Among them, two diurnal cycles (i.e., *Clusters* 1 & 3) inhibit atmospheric dispersion conditions by
enhancing atmospheric stability, suppressing PBL, and weakening surface wind speed, thus leading to air quality
deterioration on the following day.

Compared with the diurnal cycle with increasing temperature (i.e., *Cluster* 1), the atmospheric dispersion conditions are
worse for the diurnal cycle with decreasing temperature in the morning (i.e., *Cluster* 3) and cause more serious deterioration
of air quality. On the contrary, atmospheric dispersion condition with weakened atmospheric stability, deepened PBL, and
enhanced surface wind speed is obviously improved for this type of diurnal cycle with decreasing temperature in the



afternoon (i.e., *Cluster* 2), which improves the air quality on the following day. These results suggest that the identified
diurnal cycles can explain well the evolutions of atmospheric dispersion conditions and air quality on the following day. Our
findings exhibit promising potential for air quality forecasting in mountain-basin areas.

**Data availability**

The hourly air quality data, the meteorological observation data, and the ERA-5 reanalysis data were obtained from the
websites described in Sections. 2.1–2.4 and from the scientists listed in the acknowledgement. They are available from these
upon request.

**Author contributions**

DK performed data analysis, prepared the figures, and wrote original draft with contributions from all co-authors. GN
designed the research and wrote the manuscript. SW, ML, XN, and MM provided interpretation and editing of the
manuscript. JC performed data analysis and provided useful comments.

**Competing interests**

The authors declare that they have no conflict of interest.

**Acknowledgements**

This work was supported by the National Natural Science Foundation of China (91644226, 41871029, 41830648, and
41771453), the Major Scientific and Technological Projects in Sichuan Province (2018SZDZX0023), the Applied Basic
Research Project of Sichuan Science and Technology Department (2020YJ0425), the Technology Innovation Research and
Development Project of Chengdu Science and Technology Department (2018-YF05-00219-SN), the National Major Projects
on High-Resolution Earth Observation System (21-Y20B01-9001-19/22), and the appointment of M. Luo at Sun Yat-sen
University is partially supported by the Pearl River Talent Recruitment Program of Guangdong Province, China
(2017GC010634). We would like to thank the following departments for the provided data, the Ministry of Ecology and
Environment of the People's Republic of China, the China Meteorological Administration, and the European Centre for
Medium-Range Weather Forecasts.



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

**Figures**

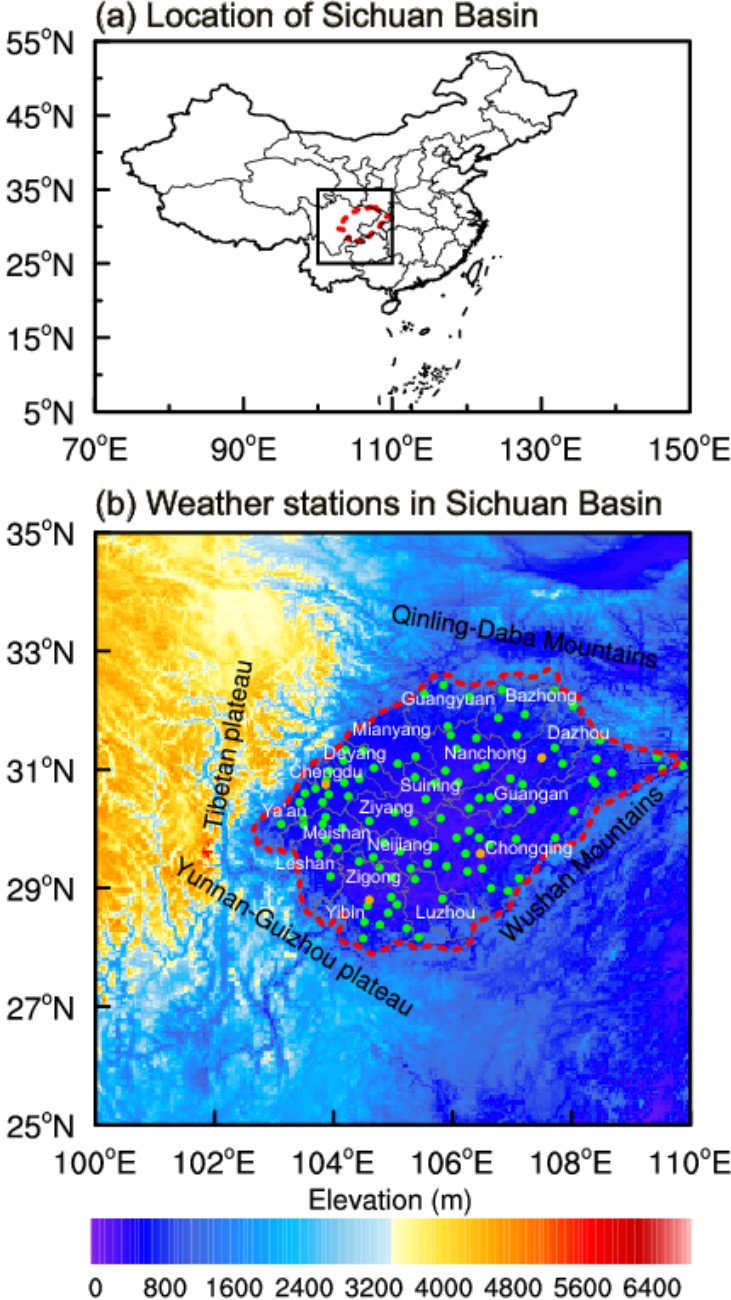


**Figure 1** Map of Sichuan Basin (SCB) in Southwest China. (**a**) Location of SCB; (**b**) Topography of SCB (shading) and the
spatial distribution of 105 meteorological stations (dots) in SCB. The dashed red line indicates the border of SCB. The
orange dots indicate the meteorological stations with radiosonde measurements.





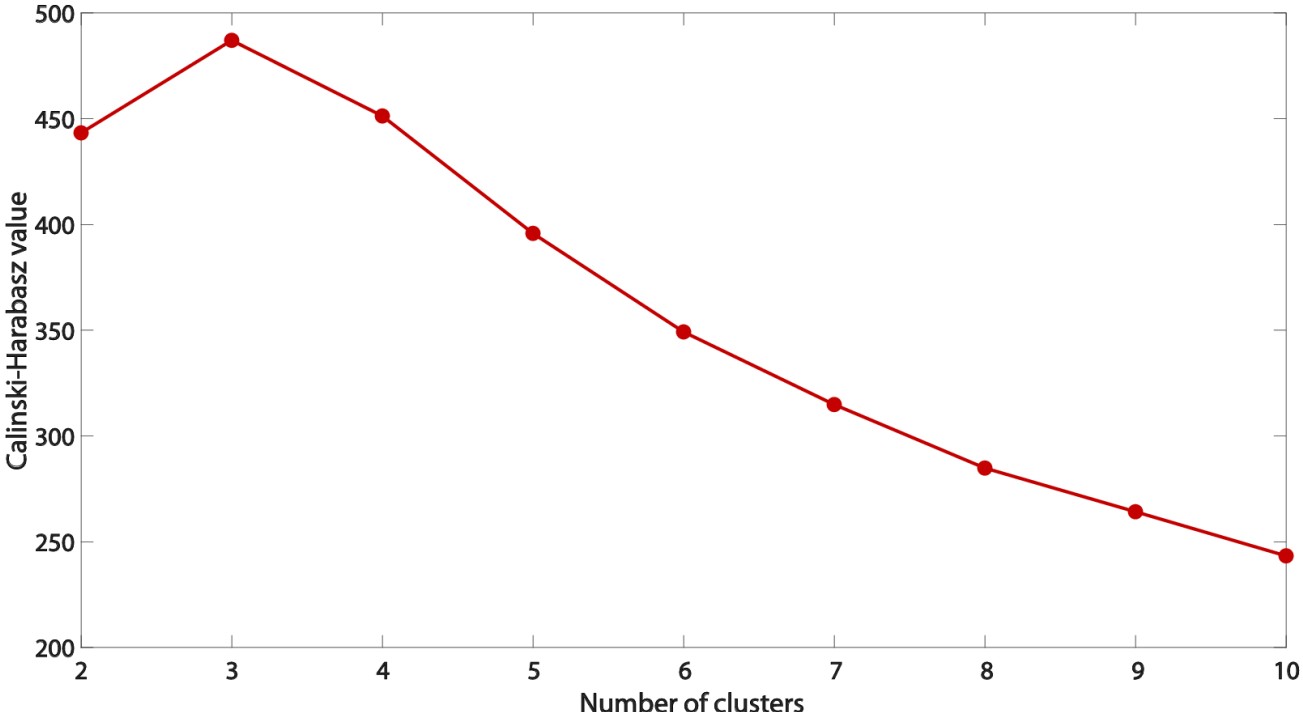


**Figure 2** Changes of Calinski-Harabasz values with different numbers of identified clusters.

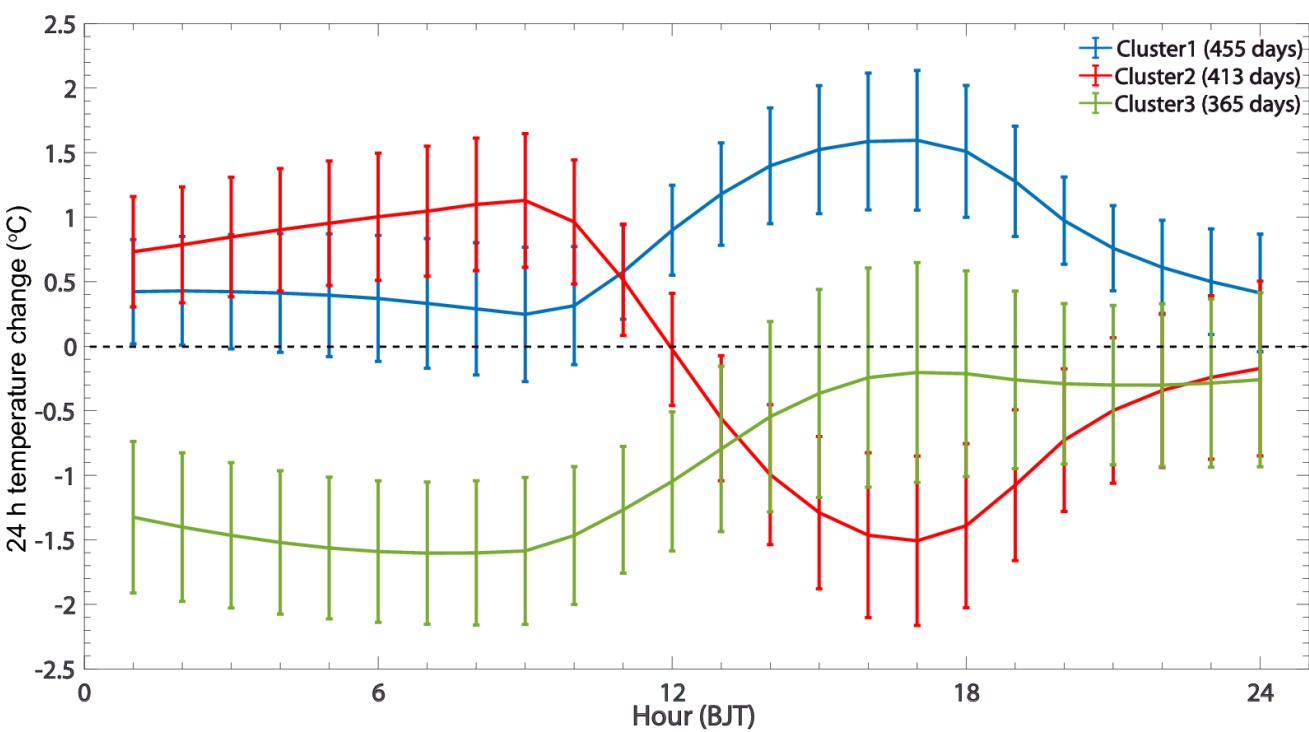


**Figure 3** Three identified diurnal cycles of day-to-day temperature change based on the K-means clustering method. The

error bar denotes the standard deviation of day-to-day temperature change.





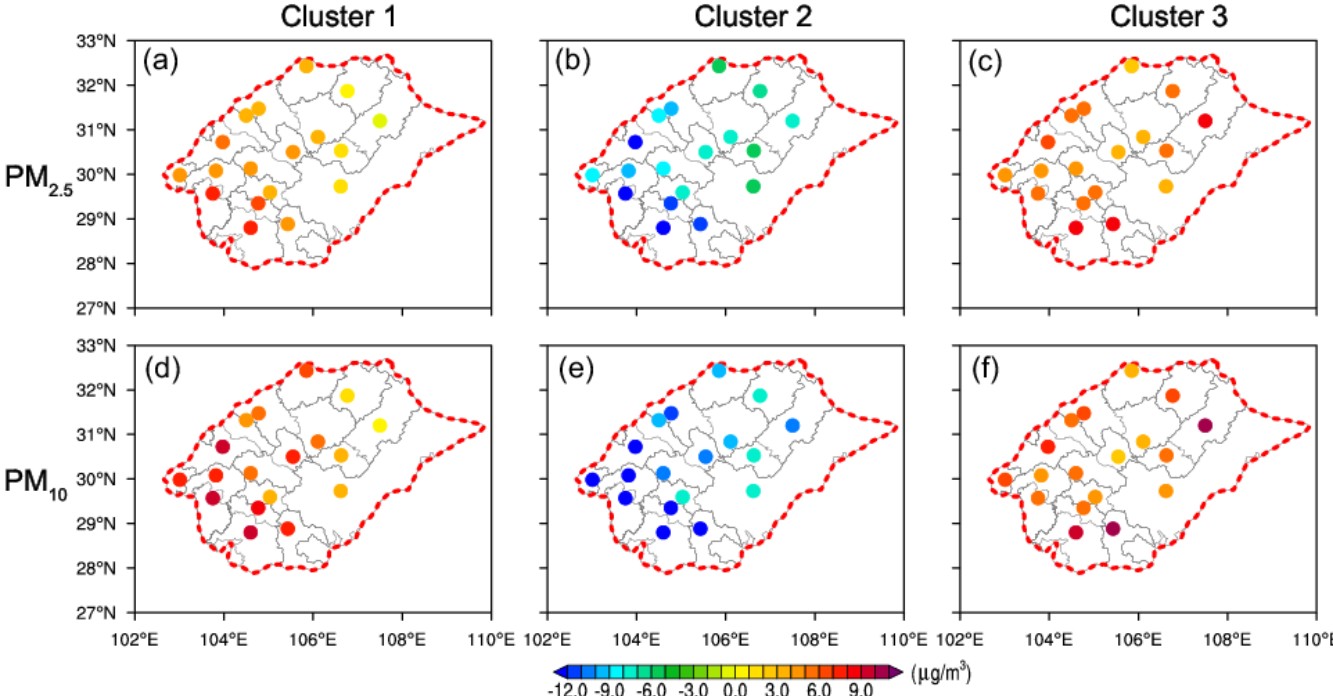

**Figure 4** Spatial distribution of the day-to-day changes in surface PM$_{2.5}$ (**a–c**) and PM$_{10}$ (**d–f**) concentrations following the three diurnal cycles within one day.



**Figure 5** Spatial distribution of the day-to-day changes in surface SO₂ (**a–c**), NO₂ (**d–f**), and CO (**g–i**) concentrations following the three identified diurnal cycles within one day.





**Figure 6** Day-to-day changes in the PT vertical profiles at 20:00 BJT following the three identified diurnal cycles within one day at four sounding stations. Chengdu (**a–c**), Yibin (**d–f**), Dazhou (**g–i**), and Chongqing (**j–l**).



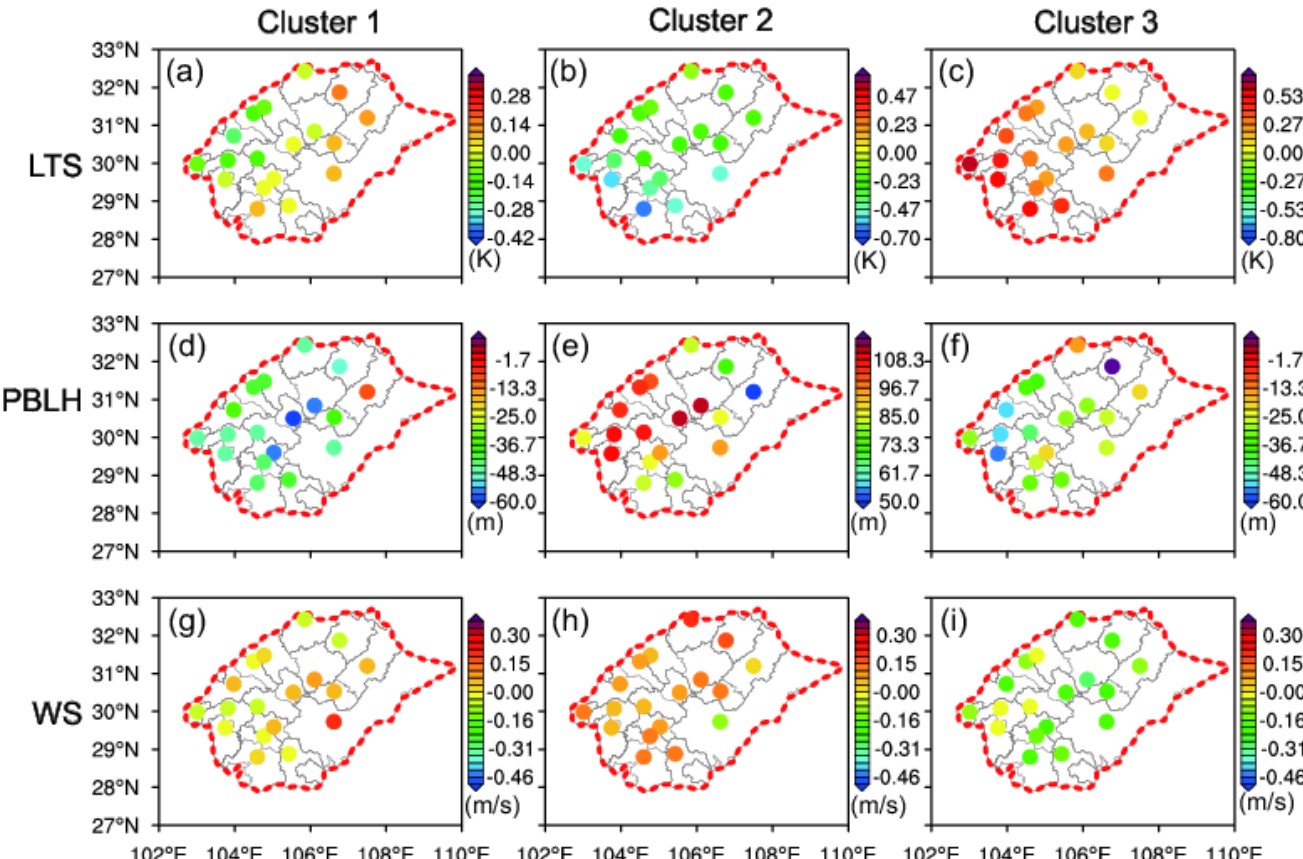

**Figure 7** Spatial distribution of the day-to-day changes in LTS (a–c), PBLH (d–f), and WS (g–i) following the three identified diurnal cycles within one day.