# Peer review of "Clustering diurnal cycles of day-to-day temperature change to understand their impacts on air quality forecasting in mountain basin areas"

_Atmospheric Chemistry and Physics, 2021_

## Referee Comment (RC2)

The manuscript by Debing Kong et al. try to explore the potential between different types of temperature diurnal cycle and air quality forecasting. The authors identify different types of diurnal cycles of day-to-day temperature change in a complex topography by taking the Sichuan Basin of China as an example. Overall, this paper is well written, and their findings are interesting and important to the community of atmospheric environment and air quality forecast. I would like to recommend an acceptation after addressing my following concerns.

1. The authors have emphasized the day-to-day variation of air quality is affected by temperature diurnal cycle, but there is no figure to show the day-to-day changes of air pollutants and meteorological conditions of hourly data.

2. The details of K-means clustering method should be specified, also the uncertainties of this method should be discussed in the section 2.5.

3. This paper shows three types of day-to-day temperature change, namely, cluster 1 (diurnal cycle with increasing temperature throughout all day), Cluster 2 (diurnal cycle with decreasing temperature in the afternoon), and Cluster 3 (diurnal cycle with decreasing temperature in the morning). The authors also showed that clusters 1 and 3 increase the air pollution in the following day while cluster 2 decreases that. Moreover, the potential physical processes related to the impacts of these clusters on air quality are also revealed. These results are interesting, but they can be further examined by including some discussion of radiation or cloud cover changes, so as to further reveal why the three types of day-to-day temperature change occur in the Sichuan Basin in winter from the view of radiation.

4. The Sichuan basin is located in the east of the Tibetan Plateau, and its weather and climate are greatly influenced by the topography. It is thus likely that topography in the Tibetan Plateau may play some roles in modulating the three types of day-to-day

temperature change. Some vertical profile of the u- and/or v-components of the wind by including the topography maybe useful to uncover the physical and dynamics reasons of the three types.

5. Are the analysis results of Sichuan Basin applicable to other heavily polluted areas? Or what is the guiding significance for the application in a wide range of areas?

Specific comments

Figure 1: does the white text indicate the name of the weather stations or the name of the major cities? Please indicate clearly.

Figure 3: please also include the percentage of the three types in this figure.

Figures 4&5: Can these figures be combined into one figure? As both of them show the changes of air pollutants associated with the three types of day-to-day temperature change.

Figure 6: does "24 h potential temperature change" mean "24 h potential change of temperature" or "24 h change of potential temperature". Please indicate it clearly. BTW, please also change "24 h" to "24-h" or "24-hour".

L48. despite -> , although

L92-93. Why not delete the data of December, 2014 to be consistent?

L95. Why 2006?

L293-294: The authors may want to indicate again clearly the data sources.

---

## Author Comment (AC1)

**Responses to Reviewer Comments**

Dear Editor and Reviewers,

We thank you very much for the valuable suggestions and comments, which are very helpful for improving the quality of our manuscript. All the comments raised by the reviewers have been addressed carefully and we prepared a list of point-by-point responses as below, and you can find our revisions in the change-tracked manuscript. Please note that reviewers' comments are in **black**, and our responses are in **blue**.

Sincerely yours,

Guicai Ning, representing all co-authors

**Reviewer #1:**

The manuscript by Debing Kong et al. provides the first attempt to examine the diurnal cycles of day-to-day temperature change and then investigates their possible impacts on winter air quality forecasting over the Sichuan Basin in China. A classification of meteorological situations is used to sort the main diurnal cycles of day-to-day temperature change occurring over this region. Three different diurnal cycles of the preceding day-to-day temperature change are identified. More interestingly, these identified diurnal cycles exhibit notably distinct effects on the evolutions of atmospheric dispersion conditions and air quality on the following day. These findings exhibit promising potential for air quality forecasting and are also critical to improve our understanding of air pollution in mountain-basin areas. The paper is well presented and logically organized. The proposed study is clear and methodologically robust. I recommend this paper to be published after these comments as follows are addressed.

*Response:* We thank you very much for your helpful comments. We have revised our manuscript carefully and prepared a list of point-by-point responses as below.

**Comments**

(1) The K-means clustering method used for classifying the diurnal cycles of day-to-day temperature change is one of the most important points in this paper. However, the methods discussion of the diurnal cycles' classification method used is too brief. For instance, the variable used in the K-means clustering is not made clear. This section needs to be much more comprehensive.

*Response:* Thanks very much for your valuable comment. According to your suggestion, the method discussion of K-means clustering has been rewritten to make this section much more comprehensive. The detailed revisions are shown as following:

Clustering methods divide the objects into specific groups, with the goal that all data objects assigned to the same cluster have common characteristics while different clusters have distinct characteristics (Darby, 2005). The clustering methods have been widely used in climate and environmental researches (Bardossy et al., 1995; Cavazos, 2000; Luo and Lau, 2017; Bernier et al., 2019). In this study, the regional average values of day-to-day temperature change in SCB and the K-means clustering method (MacQueen, 1967) are selected to classify the diurnal cycles of day-to-day temperature change, because of the simplicity and convergence characteristics of K-means clustering method. The details of K-means clustering method can refer to MacQueen (1967) and (Mokdad and Haddad, 2017) and is also provided in the **supplementary document**.

K-means is one of the most commonly used unsupervised learning algorithms that treat the renowned clustering problem (MacQueen, 1967; Hartigan and Wong, 1979; Mokdad and Haddad, 2017). This is, by automatically partitioning the given data set into a certain number of groups selected a priori (assume k clusters). The aim of the K-means algorithm is to divide M points in N dimensions into K clusters so that the within-cluster sum of squares is minimized. Then, the initial cluster centers are iteratively refined as follows.

Each data point is assigned to its neighboring cluster centroid based on the Euclidean distance metric.

Each cluster centroid is then re-calculated to be the mean of its constituent data points. This can be achieved by minimizing an objective function known as a squared error function. It is defined as:

$$J(v) = \sum_{i=1}^{k} \sum_{j=1}^{c_i} (||x_i - v_j||)^2$$

where

$k$: is the number of cluster centers;

$c_i$: is the number of data points in the $i^{th}$ cluster;

$||x_i-v_j||$: is the Euclidean distance between $x_i$ and $v_j$;

$v_j$: is the data points in the $i^{th}$ cluster;

$x_i$: is the centroid vector of the $i^{th}$ cluster.

When there is no further change in assignment of data point to clusters, the K-means algorithm converges to the optimal solution.

(2) In this paper, the authors found that the three different diurnal cycles of the preceding day-to-day temperature change exhibit notably distinct effects on the evolutions of air pollutants' concentrations on the following day. However, Figure 4&5 only depict the spatial distributions of the following day-to-day changes in absolute concentrations of particulate pollutants and gaseous pollutants. To exhibit the change range of air pollutants' concentrations on the following day more intuitively, I suggest the authors to add some investigations about the percentage values of the changes in air pollutants' concentrations.

*Response:* Thanks very much for your valuable comment. According to your comment, the percentage values of the air quality changes are also investigated and are shown as the follow figure. Moreover, the descriptions of the percentage values are also added in the revised manuscript.

[Figure]

**Figure S1** Spatial distribution of percentage values of the day-to-day changes in surface $PM_{2.5}$ (**a–c**), $PM_{10}$ (**d–f**), $SO_2$ (**g–i**), $NO_2$ (**j–l**), and CO (**m–o**) concentrations following the three identified diurnal cycles within one day.

(3) By using K-means clustering method, three dominant diurnal cycles of day-to-day temperature change are identified in Sichuan Basin. The basic features of the three diurnal cycles are shown in Figure 3, Cluster 1 exhibits diurnal cycle with increasing temperature throughout all day, Cluster 2 shows diurnal cycle with decreasing temperature in the afternoon, and Cluster 3 exhibits diurnal cycle with decreasing temperature in the morning. As shown in Figure 3, the diurnal distributions of temperature changes corresponding to different clusters are very different, which also pose notably effects on the atmospheric

dispersion conditions and air quality. In general, atmospheric radiation and temperature advection are the important factors leading to changes in air temperature. In particular, atmospheric radiation could play a key role in resulting the different features in temperature changes between daytime and nighttime. Thus, the authors should investigate the behaviors of cloud cover (including low cloud cover and total cloud cover) to reveal the possible causes inducing the above three dominant diurnal cycles of day-to-day temperature change.

*Response:* Thanks very much for your valuable comment. In the revision, to reveal the underlying mechanism of the formation of the above three diurnal cycles of day-to-day temperature change, we also investigate the nighttime and daytime day-to-day changes in total cloud cover that could play a key role in temperature changes by modulating atmospheric radiations. **Figure 4** shows the nighttime and daytime day-to-day changes in total cloud cover associated with the three diurnal cycles. Corresponding to the diurnal cycle with increasing temperature (*Cluster* 1), the total cloud exhibits slightly increase in the eastern of SCB, while decrease in the western of SCB (**Figure 4a**). The dipole spatial distribution could result in a weak changes in the regional average temperature across SCB during nighttime (**Figure 3**). During daytime, negative changes in total cloud cover are observed in the entire basin (**Figure 4d**) that are beneficial to the obviously increasing in temperature in the afternoon (**Figure 3**). On the contrary, both the nighttime and daytime changes in total cloud cover are positive in the entire basin for *Cluster* 2 (**Figure 4b and e**), which could induce the increasing temperature during nighttime and decreasing temperature during afternoon (**Figure 3**). Corresponding to the diurnal cycle with decreasing temperature in the morning (*Cluster* 3), obviously decreasing in the total cloud cover are observed in the entire basin during nighttime (**Figure 4c**) that are beneficial to the temperature decreasing.

[Figure]

**Figure 4** The nighttime (**a-c**) and daytime (**d-f**) day-to-day changes in total cloud cover associated with the three diurnal cycles.

(4) The blank space on the right side of Figure 1a is too large. I suggest the authors to adjust the X coordinate axis of Figure 1a to 70 °E~140 °E.

*Response:* The updated figure is shown as below.

[Figure]

**Figure 1** Map of Sichuan Basin (SCB) in Southwest China. (a) Location of SCB; (b) Topography of SCB (shading) and the spatial distribution of 105 meteorological stations (dots) in SCB. The dashed red line indicates the border of SCB. The orange dots indicate the meteorological stations with radiosonde measurements. The white text indicate the name of the major cities in SCB.

(5) Line 211 show higher temperature change in the level between middle level (800-850 hPa) than the lower level (900-950 hPa) -> show higher temperature change in the higher level (800-850 hPa) than the lower level (900-950 hPa).
*Response:* Corrected.

(6) Line 22: for the first time we -> for the first time, we
*Response:* Corrected.

(7) Lines 56-58: The key questions include … -> There are two key questions. The first one is what are the behaviors of ... and the second one is how these behaviors affect air quality ...
*Response:* Corrected.

(8) Line 59: understanding of winter air pollution -> understanding winter air pollution
*Response:* Corrected.

(9) Line 62: local residents -> residents
*Response:* Corrected.

(10) Line 69: conditions -> conditions'
*Response:* Corrected.

(11) Line 77: Our study is expected to -> We expect our study to
*Response:* Corrected.

(12) Line 91: since -> on
*Response:* Corrected.

(13) Line 113: can be used to evaluate -> can evaluate
*Response:* Corrected.

(14) Line 190: showed -> shown

*Response:* Corrected.

(15) Line 237: playing key role -> playing a key role

*Response:* Corrected.

**References**

Bardossy, A., Duckstein, L., and Bogardi, I.: Fuzzy rule-based classification of atmospheric circulation patterns, Int. J. Climatol., 15, 1087-1097, doi: 10.1002/joc.3370151003, 1995.

Bernier, C., Wang, Y., Estes, M., Lei, R., Jia, B., Wang, S.-C., and Sun, J.: Clustering surface ozone diurnal cycles to understand the impact of circulation patterns in Houston, TX, J. Geophys. Res. Atmos., 124, 13457-13474, doi: 10.1029/2019JD031725, 2019.

Cavazos, T.: Using self-organizing maps to investigate extreme climate events: an application to wintertime precipitation in the Balkans, J. Clim., 13, 1718-1732, doi: 10.1175/1520-0442(2000)013<1718:USOMTI>2.0.CO;2, 2000.

Darby, L. S.: Cluster analysis of surface winds in Houston, Texas, and the impact of wind patterns on ozone, J. Appl. Meteorol. Climatol., 44, 1788-1806, doi: 10.1175/JAM2320.1, 2005.

Hartigan, J. A., and Wong, M. A.: Algorithm AS 136: A K-Means Clustering Algorithm, Journal of the Royal Statistical Society. Series C (Applied Statistics), 28, 100-108, doi: 10.2307/2346830, 1979.

Luo, M., and Lau, N.-C.: Heat waves in southern China: synoptic behavior, long-term change, and urbanization effects, J. Clim., 30, 703-720, doi: 10.1175/JCLI-D-16-0269.1, 2017.

MacQueen, J.: Some methods for classification and analysis of multivariate observations, Proceedings of the fifth Berkeley symposium on mathematical statistics and probability, 1967, 281-297.

Mokdad, F., and Haddad, B.: Improved infrared precipitation estimation approaches based on k-means clustering: application to north Algeria using MSG-SEVIRI satellite data, Adv. Space Res., 59, 2880-2900, doi: 10.1016/j.asr.2017.03.027, 2017.

---

## Author Comment (AC2)

**Responses to Reviewer Comments**

Dear Editor and Reviewers,

We thank you very much for the valuable suggestions and comments, which are very helpful for improving the quality of our manuscript. All the comments raised by the reviewers have been addressed carefully and we prepared a list of point-by-point responses as below, and you can find our revisions in the change-tracked manuscript. Please note that reviewers' comments are in **black**, and our responses are in **blue**.

Sincerely yours,
Guicai Ning, representing all co-authors

**Reviewer #2:**

The manuscript by Debing Kong et al. try to explore the potential between different types of temperature diurnal cycle and air quality forecasting. The authors identify different types of diurnal cycles of day-to-day temperature change in a complex topography by taking the Sichuan Basin of China as an example. Overall, this paper is well written, and their findings are interesting and important to the community of atmospheric environment and air quality forecast. I would like to recommend an acceptation after addressing my following concerns.

*Response:* We thank you very much for your helpful comments. We have revised our manuscript carefully and prepared a list of point-by-point responses as below.

(1) The authors have emphasized the day-to-day variation of air quality is affected by temperature diurnal cycle, but there is no figure to show the day-to-day changes of air pollutants and meteorological conditions of hourly data.

*Response:* We thank you very much for your valuable comments. In this study, we use K-means clustering method and the observed hourly temperature data to classify the diurnal cycles of day-to-day temperature change and then to investigate the impacts of these

diurnal cycles on the following day-to-day changes in daily mean air quality. The hourly temperature features can be found in **Figure 3** that is shown as below.

[Figure]

**Figure 3** Three identified diurnal cycles of day-to-day temperature change based on the K-means clustering method. The error bar denotes the standard deviation of day-to-day temperature change.

(2) The details of K-means clustering method should be specified, also the uncertainties of this method should be discussed in the section 2.5.

*Response:* We thank you very much for your valuable comments. The details and the uncertainties of K-means clustering method are specified as below.

Clustering methods divide the objects into specific groups, with the goal that all data objects assigned to the same cluster have common characteristics while different clusters have distinct characteristics (Darby, 2005). The clustering methods have been widely used in climate and environmental researches (Bardossy et al., 1995; Cavazos, 2000; Luo and Lau, 2017; Bernier et al., 2019). In this study, the regional average values of day-to-day temperature change in SCB and the K-means clustering method (MacQueen, 1967) are selected to classify the diurnal cycles of day-to-day temperature change, because of the simplicity and convergence characteristics of K-means clustering method. The details of K-means clustering method can refer to MacQueen (1967) and (Mokdad and Haddad, 2017) and is also provided in the **supplementary document**.

K-means is one of the most commonly used unsupervised learning algorithms that treat the renowned clustering problem (MacQueen, 1967; Hartigan and Wong, 1979; Mokdad and

Haddad, 2017). This is, by automatically partitioning the given data set into a certain number of groups selected a priori (assume k clusters). The aim of the K-means algorithm is to divide M points in N dimensions into K clusters so that the within-cluster sum of squares is minimized. Then, the initial cluster centers are iteratively refined as follows. Each data point is assigned to its neighboring cluster centroid based on the Euclidean distance metric.

Each cluster centroid is then re-calculated to be the mean of its constituent data points. This can be achieved by minimizing an objective function known as a squared error function. It is defined as:

$$J(v) = \sum_{i=1}^{k} \sum_{j=1}^{c_i} (||x_i - v_j||)^2$$

where

$k$: is the number of cluster centers;

$c_i$: is the number of data points in the $i^{th}$ cluster;

$||x_i-v_j||$: is the Euclidean distance between $x_i$ and $v_j$;

$v_j$: is the data points in the $i^{th}$ cluster;

$x_i$: is the centroid vector of the $i^{th}$ cluster.

When there is no further change in assignment of data point to clusters, the K-means algorithm converges to the optimal solution.

(3) This paper shows three types of day-to-day temperature change, namely, cluster 1 (diurnal cycle with increasing temperature throughout all day), Cluster 2 (diurnal cycle with decreasing temperature in the afternoon), and Cluster 3 (diurnal cycle with decreasing temperature in the morning). The authors also showed that clusters 1 and 3 increase the air pollution in the following day while cluster 2 decreases that. Moreover, the potential physical processes related to the impacts of these clusters on air quality are also revealed. These results are interesting, but they can be further examined by including some discussion of radiation or cloud cover changes, so as to further reveal why the three types of day-to-day temperature change occur in the Sichuan Basin in winter from the view of radiation.

*Response:* Thanks very much for your valuable comment. To reveal why the three diurnal cycles of day-to-day temperature change occur in SCB in winter from the view of radiation, we also investigate the nighttime and daytime day-to-day changes in total cloud cover that could play a key role in temperature changes by modulating atmospheric radiations.

**Figure 4** shows the nighttime and daytime day-to-day changes in total cloud cover associated with the three diurnal cycles. Corresponding to the diurnal cycle with increasing temperature (*Cluster* 1), the total cloud exhibits slightly increase in the eastern of SCB, while decrease in the western of SCB (**Figure 4a**). The dipole spatial distribution could result in a weak changes in the regional average temperature across SCB during nighttime (**Figure 3**). During daytime, negative changes in total cloud cover are observed in the entire basin (**Figure 4d**) that are beneficial to the obviously increasing in temperature in the afternoon (**Figure 3**). On the contrary, both the nighttime and daytime changes in total cloud cover are positive in the entire basin for *Cluster* 2 (**Figure 4b and e**), which could induce the increasing temperature during nighttime and decreasing temperature during afternoon (**Figure 3**). Corresponding to the diurnal cycle with decreasing temperature in the morning (*Cluster* 3), obviously decreasing in the total cloud cover are observed in the entire basin during nighttime (**Figure 4c**) that are beneficial to the temperature decreasing.

[Figure]

**Figure 4** The nighttime (**a-c**) and daytime (**d-f**) day-to-day changes in total cloud cover associated with the three diurnal cycles.

(4) The Sichuan basin is located in the east of the Tibetan Plateau, and its weather and climate are greatly influenced by the topography. It is thus likely that topography in the Tibetan Plateau may play some roles in modulating the three types of day-to-day temperature change. Some vertical profile of the u- and/or v-components of the wind by including the topography maybe useful to uncover the physical and dynamics reasons of the three types.

*Response:* Thanks very much for your valuable comment. In the revision, the vertical west–east cross-sections of the day-to-day changes in wind vectors (synthesized by *u* and *w*) at 14:00 BJT are investigated to uncover the physical and dynamics reasons of the formation of the above diurnal cycles of day-to-day temperature change. The details are shown as below.

Moreover, SCB is located in the eastern Tibetan Plateau and the complex topography could play the key role in modulating the temperature changes over SCB (Ning et al., 2018; Ning et al., 2019). Therefore, the vertical west–east cross-sections of the day-to-day changes in wind vectors (synthesized by *u* and *w*) at 14:00 BJT are also investigated to uncover the physical and dynamics reasons of the formation of the above diurnal cycles of day-to-day temperature change. As shown in **Figure 5b**, a significantly ascending motion is observed over SCB that could induce the obviously decreasing temperature in the afternoon for *Cluster* 2 (**Figure 3**). On the contrary, the descending motion prevails over SCB for *Cluster* 1 and *Cluster* 3, which is beneficial to the temperature increasing in the afternoon and thus plays a key role in the day-to-day temperature change for these two diurnal cycles.

[Figure]

**Figure 5** Vertical west–east cross-sections of the day-to-day changes in wind vectors (synthesized by $u$ and $w$) at 14:00 BJT through the SCB (30.75°N) associated with the three diurnal cycles. Note that the vertical velocity is multiplied by -50 when plotting the wind vectors. The units for $u$ and $w$ are m/s and Pa/s, respectively. The complex terrain is marked by grey shading.

(5) Are the analysis results of Sichuan Basin applicable to other heavily polluted areas? Or what is the guiding significance for the application in a wide range of areas?

*Response:* Thanks very much for your valuable comment. In this study, we take Sichuan Basin as an example to cluster diurnal cycles of day-to-day temperature change and then to investigate their impacts on air quality forecasting in mountain-basin areas.

It is noted that Sichuan Basin has a unique topography, with Qinling-Daba and Wu mountains in the north and east and with Qinghai-Tibet Plateau and Yunnan-Guizhou Plateau in the west and south of the basin (**Fig. 1**). The combination of these complex topography results in unique weather and climate, like the southwest vortex and the Huaxi Autumn rain season etc. The southwest vortex, southern branch, and Qinghai-Tibet high pressure are often formed over Sichuan Basin or Tibetan plateau and the complex synoptic systems significantly affect atmospheric dispersion conditions (Wang et al., 1993; Wei et al., 2014; Feng et al., 2016; Yu et al., 2016; Ning et al., 2019; Ning et al., 2020). Moreover, cold air is difficult to directly invade into Sichuan Basin as the basin is blocked by the surrounding mountains and the temperature changes over Sichuan Basin are significantly modulated by the Tibetan plateau (Ning et al., 2018; Ning et al., 2019). Therefore, both the physical mechanism of atmospheric conditions' effects on air pollution and the air quality forecasting in SCB are more complicated than these in the eastern plain regions of China (Chen and Xie, 2012; Wang et al., 2014; Ning et al., 2019; Zhang et al., 2019).

Thus, the analysis results of Sichuan Basin are applicable to other heavily polluted areas with complex topography, such as mountain-basin areas, which are critical to improve the understanding of air pollution and exhibit promising potential for air quality forecasting in mountain-basin areas.

**Specific comments**

Figure 1: does the white text indicate the name of the weather stations or the name of the major cities? Please indicate clearly.

*Response:* The white text in **Figure 1** indicate the name of the major cities in Sichuan Basin, which are added in the updated **Figure 1**.

Figure 3: please also include the percentage of the three types in this figure.

*Response:* The percentage of the three types are added in the updated **Figure 3**.

Figures 4&5: Can these figures be combined into one figure? As both of them show the changes of air pollutants associated with the three types of day-to-day temperature change.

*Response:* According to your comment, Figures 4&5 are combined into one figure that is shown as below.

[Figure]

**Figure 6** Spatial distribution of the day-to-day changes in surface PM$_{2.5}$ (**a–c**), PM$_{10}$ (**d–f**), SO$_2$ (**g–i**), NO$_2$ (**j–l**), and CO (**m–o**) concentrations following the three diurnal cycles within one day.

Figure 6: does "24 h potential temperature change" mean "24 h potential change of temperature" or "24 h change of potential temperature". Please indicate it clearly. BTW, please also change "24 h" to "24-h" or "24-hour".

*Response:* "24 h potential temperature change" means "24-h change of potential temperature" and has been revised according to your comment.

L48. despite -> , although

*Response:* Corrected.

L92-93. Why not delete the data of December, 2014 to be consistent?

*Response:* To enhance the robust of our results, we want to get more air quality data samples as more as possible. We thus have no delete the data of December, 2014.

L95. Why 2006?

*Response:* To get more stable and reliable diurnal cycles of day-to-day temperature changes in Sichuan Basin by K-means clustering method, we thus collect the hourly winter surface temperature data from December 2006 when the China Meteorological Administration began to provide hourly meteorological data to get more data samples as more as possible.

L293-294: The authors may want to indicate again clearly the data sources.

*Response:* Thanks very much for your valuable comment. The details of the data sources are added in the revision and are shown as below.

[revised manuscript text omitted]